# Biochemical and Rapid Molecular Analyses to Identify Glyphosate Resistance in *Lolium* spp.

Maria Gerakari [1], Nikolina Cheimona [2], Eleni Tani [1,*], Ilias Travlos [3], Demosthenis Chachalis [2], Donato Loddo [4], Solvejg Kopp Mathiassen [5], Thomas K. Gitsopoulos [6], Laura Scarabel [4], Silvia Panozzo [4], Michael Kristensen [5], Per Kudsk [5] and Maurizio Sattin [4]

1   Laboratory of Plant Breeding and Biometry, Agricultural University of Athens, 11855 Athens, Greece; mgerakari@aua.gr
2   Laboratory of Weed Science, Benaki Phytopathological Institute, 14561 Kifisia, Greece; n.cheimona@bpi.gr (N.C.); d.chachalis@bpi.gr (D.C.)
3   Laboratory of Agronomy, Agricultural University of Athens, 11855 Athens, Greece; travlos@aua.gr
4   Institute for Sustainable Plant Protection (IPSP-CNR), 35020 Legnaro, Italy; donato.loddo@cnr.it (D.L.); laura.scarabel@cnr.it (L.S.); silvia.panozzo@ipsp.cnr.it (S.P.); maurizio.sattin@cnr.it (M.S.)
5   Department of Agroecology, Aarhus University, DK-4200 Slagelse, Denmark; sma@agro.au.dk (S.K.M.); mikr@agro.au.dk (M.K.); per.kudsk@agro.au.dk (P.K.)
6   HAO-DEMETER, Institute of Plant Breeding and Genetic Resources, 57001 Thermi, Greece; gitsopoulos@ipgrb.gr
*   Correspondence: etani@aua.gr

**Abstract:** *Lolium* spp. are troublesome weeds mainly found in winter cereal crops worldwide, including Europe. In recent years resistant mechanisms have been evolved to several important herbicides. In this study we investigated the mechanisms responsible for conferring glyphosate resistance in some *Lolium* spp. populations. A holistic approach was used, based on dose-response experiments, determination of shikimic acid concentration in plant leaf tissue, as well as molecular analyses. More specifically, in three *Lolium* spp. populations the existence of a mutation in the Pro-106 codon of the 5-enolpyruvylshikimate-3 phosphate synthase (*EPSPS*) gene was investigated as well as the relative transcript levels of four *ABC-transporter* genes were monitored at three time points after glyphosate application. The results demonstrated that glyphosate resistance is a multifactor phenomenon. Relative transcript levels of the *ABC-transporter* genes were abundant at very early time points after glyphosate treatments. Dose-response experiments and shikimate analyses were in accordance with the findings of the quantitative PCR (qPCR) analyses. We suggest that relative expression ratio of *ABC-transporter* genes can be a useful tool to rapidly identify *Lolium* spp. populations resistant to glyphosate.

**Keywords:** ryegrass resistance mechanisms; dose-response; shikimic acid; *EPSPS* gene; *ABC-transporters*; TSR; NTSR

## 1. Introduction

Weeds are the most important biotic factors affecting agricultural production and causing crop yield losses worldwide. *Lolium rigidum* Gaud. (rigid ryegrass) and *Lolium multiflorum* Lam. are annual, cross-pollinated grass weeds which cause severe problems mainly in winter cereal crops and to a lesser extent in other winter and early spring crops. The effective control of these weeds is crucial for crop productivity, as they have evolved resistance towards some of the most widely applied herbicides, including glyphosate [1–4]. Glyphosate is the most used, broad-spectrum herbicide that has been used for more than 50 years. It acts as a competitive inhibitor of the enzyme 5-enolpyruvylshikimate 3′-phosphate synthase (EPSPS), which is a key enzyme in the shikimic acid pathway that catalyzes the synthesis of 5′ enoylpyruvylshikimate-3-phosphate (EPSP) from shikimate-3-phosphate (S3P) and phosphoenoylpyruvate (PEP) [5].

Over reliance on glyphosate and its repeated use have led to the occurrence of several glyphosate-resistant weed species. Glyphosate resistance was first reported in a *L. rigidum* population being exposed for many years on two–three applications per year [6]. Resistance to glyphosate has evolved mostly in the genetically diverse and resistance-prone genera *Amaranthus*, *Conyza* and *Lolium*, in situations with intense glyphosate selection pressure [7]. Currently, there are 338 reported cases of glyphosate-resistant weeds affecting 51 species [8].

Biochemical screening tests (e.g., shikimate analysis) are widely used for documenting glyphosate resistance beyond the classic dose–response experiments [9]. Inhibition of EPSPS triggers the accumulation of shikimate, which is a substrate of the enzyme. Therefore, the effect of glyphosate on plants can be determined by monitoring the accumulation of shikimate [10].

Resistance mechanisms to glyphosate can be generally classified as target-site based (TSR) and non-target site based (NTSR) [11]. Studies indicate that target-site resistance is endowed by a mutation in the key herbicide target gene (*EPSPS*) [12]. Multiple missense mutations, which result in substitution of the proline residue at codon 106 of the target enzyme EPSPS, have been identified in glyphosate-resistant individuals of various weed species, including *L. multiflorum* and *L. rigidum* [13–16]. The NTSR mechanisms include reduced herbicide uptake and translocation [17,18], sequestration in vacuoles [19] and rapid degradation of the herbicide to non-harmful compounds [17,20]. The availability of high-throughput transcriptome analyses has recently facilitated the understanding of NTSR mechanisms in some plant species [21–23]. Several studies have identified *ABC transporters* as herbicide-resistant gene candidates that mainly sequester the herbicide into vacuoles [24–26]. Tani et al. [27] have reported that glyphosate resistance (especially to high doses) in *C. canadensis* was attributed to high induction of *ABC transporter* genes *M10* and *M11*. Glyphosate-resistant populations of *Echinochloa colona* overexpressed an *ABC transporter* gene (*EcABCC8*). When expressed in transgenic rice, this *EcABCC8* transporter endowed glyphosate resistance. Similarly, when rice, maize, and soybean overexpressed the *EcABCC8* ortholog genes they displayed resistance to glyphosate [28]. Additionally, researchers demonstrated that *ABC-transporters* and multidrug proteins are involved in several physiological processes and may be associated with herbicide resistance mechanisms involving impaired glyphosate herbicide translocation through mediating active secretions of chemicals from roots [29]. Cechin et al. [30] recently presented a list of 21 candidate genes (including *ABC-transporters*) that may be involved in the NTSR to glyphosate in *L. multiflorum*.

This study aims to investigate the TRS mechanisms by *EPSPS* gene sequencing mainly focusing on the region covering the Pro 106 codon. Moreover, qPCR analysis at three different time points after glyphosate application was employed to identify *ABC transporters* candidate genes involved in NTSR in resistant populations. The early detection of high relative transcripts could serve as an easy and reliable tool for detecting glyphosate resistance in *Lolium* spp. populations shortly after glyphosate application.

## 2. Materials and Methods

### 2.1. Plant Material

*L. rigidum* seeds were collected in two different Greek regions where *L. rigidum* was not poorly controlled by glyphosate applied at the recommended field rate (720 g ae ha$^{-1}$). Similarly, *L. rigidum* seeds were collected from a vineyard field in central Greece where glyphosate had not been used before (Table 1). In addition, a *L. multiflorum* population, previously reported as glyphosate-resistant, and collected in a field managed with conservation agriculture in north-eastern Italy was included in the study (pop.4). Seeds from 20 plants per each population were collected in each field. To break dormancy, seeds were vernalized in a refrigerator at 4 °C in Petri dishes on wet filter paper, in dark conditions for 4 days. The seeds were then, placed in a germination chamber and kept for 5 days under a temperature regime (day/night) of 25/15 °C and a 12 h photoperiod of artificial light. Ten seedlings at a similar growth stage were transplanted into pots (15 × 15 × 20 cm) filled with

a standard potting mix (50% silty loam soil, 25% perlite and 25% peat). The experiment was conducted in the period from December 2019—February 2020 in a greenhouse with temperatures ranging from 12 to 28 °C and the pots were watered regularly to maintain the growth medium at field capacity.

**Table 1.** Population code, infested crop, and area of *Lolium* spp. populations included in the study.

| Population Code | Crop | Area |
|---|---|---|
| pop.1 | Olives | western Greece |
| pop.2 | barley | central Greece |
| pop.3 | Vineyard | central Greece |
| pop.4 | cereal stubble in no-till fields | Italy |

### 2.2. Dose-Response Experiment

Glyphosate was applied to plants at BBCH 14–21 using a custom-built, compressed-air, low-pressure experimental sprayer fitted with flat-fan nozzles calibrated to deliver 300 L ha$^{-1}$ at 250 kPa. The following herbicide doses (expressed as x-fold of label recommended field rate = 720 g ae ha$^{-1}$) were applied: $1/16\times$ (only for the potentially susceptible populations), $1/8\times, 1/4\times, 1/2\times, 1\times, 2\times, 4\times, 8\times$ (the latter only for the potentially resistant populations). The pots were arranged in a completely randomized block design with 4 replicates.

Fresh weight reduction and survival of *Lolium* spp. plants were evaluated at 4 weeks after the treatment (WAT). Plant fresh biomass was recorded for each pot (replicate), the above-ground shoots were collected and weighed, including those from dead plants. Fresh biomass was expressed as percentage (%) compared to the untreated control. The growth rates (GRs) causing 50% and 90% reduction in fresh weight ($GR_{50}$-$GR_{90}$), were estimated by non-linear regression using the following log-logistic equation [31]:

$$y = c + < (d - c)/1 + \exp \{b [\log (x) - \log (GR_{50/90})]\}>$$

where y represents dry weight at herbicide dose (x) whereas c and d denote the lower and upper limits, respectively, $GR_{50/90}$ is the herbicide dose centered between the asymptotic values, and b is the slope of the response curve.

The resistance index (RI) was calculated as the ratio of the $GR_{90}$ of the considered resistant populations (pop.1, 3, 4) to the $GR_{90}$ of the pop.2 regarded as the susceptible population of the study (Table 1).

### 2.3. Shikimate Measurement

Shikimic acid concentration in plant tissue of the four populations of *Lolium* spp. collected 72 h after glyphosate treatment (at the recommended dose) was determined to further evaluate the level of resistance to glyphosate using the method of Tani at al. [27] with some modifications. Some populations at the present research study have been previously studied and confirmed as glyphosate resistant [32].

The shikimate acid was determined for a bulk sample of 3 different plants in 3 replicates. Briefly, shikimic acid measurement was conducted by using hydrochloride HCl. For each sample, 0.1 g of young leaves were ground with 1 mL of HCl and left at room temperature for 24 h. After the shikimic acid isolation stage, the samples were centrifuged for 1 min at 3000 rpm and 75 µL of supernatant was transferred in a new tube. For the oxidation stage, deionized water and 500 µL of oxidation solution were added in each tube to a final volume of 1 mL. The tubes were left at room temperature for 3 h. Finally, for the chromophore stage, 300 µL solution from each tube was transferred to a new one and 700 µL of deionized water, 400 µL of sodium sulfate and 600 µL NaOH were added. Finally, the absorbance at 380 nm was determined with a spectrophotometer for each sample.

### 2.4. RNA Isolation and EPSPS Target-Site Resistance Analyses

The plant material used for molecular analyses was derived from populations 1, 3 and 4, these considered to be resistant based on the results of dose-response and shikimate analyses. A sample consisting of material from three different plants were collected per three replications at three different time points, three, six and 12 h after spraying with the recommended dose of glyphosate (720 g ae ha$^{-1}$). Plants without any treatment (controls) were also collected and analyzed. RNA was isolated from leaves using NucleoZol solution according to the manufacturer's protocol. First-strand cDNA was synthesized using 0.5 g of total RNA and Primescript RT enzyme according to the manufacturer's instructions (Takara, Tokyo, Japan). The cDNAs were diluted to 100 μL with sterile water. The selected primers had been previously used for detecting mutations at P106 codon in *EPSPS* gene of *Lolium rigidum* [33]. The forward primer was *EPSPS-F*: 5′ TCTTCTTGGGGAACGCTGGA 3′ and the reverse was *EPSPS-R*: 5′ TAACCTTGCCACCAGGTAGCCCTC 3′. PCR reactions contained: 3 μL of 50 ng concentration cDNA sample and 22 μL Master Mix containing the following: 2 mM MgCl$_2$, 0.2 mM dNTPs, 0.2 mM from each primer, 1× Buffer with Mg, 0.625 U polymerase enzyme and RNase free H$_2$O to the final volume of 25 μL. The PCR program used was as follows: 1 cycle at 95 °C for 5 min, 40 cycles with the following stages: 95 °C for 30 s; 60 °C for 30 s; 72 °C for 1m and a last cycle at 72 °C for 10 min. PCR products were sequenced and the consensus sequences for each population were assembled using Clustal Omega (Multiple Sequence Alignment) (https://www.ebi.ac.uk/Tools/msa/clustalo/ accessed on 1 November 2021).

### 2.5. Real-Time Polymerase Chain Reaction (PCR) Experiments

Relative expression of four *ABC-transporter* genes reported from [34] (*ABC III*, JZ166942.1, *Lpmultidrugprotein* JF747403.1, *Lproninhibited ABC* JF747419.1) and *LpM10* (retrieved from NCBI datadase with homology to *Conyza M10* gene [27]) were studied (3, 6 and 12 h after glyphosate application). As a reference gene cinnamoyl-CoA reductase (CCR) was used. CCR is constitutively expressed and is present as a single copy gene in perennial ryegrass [35]. Populations 1, 3 and 4 were used for this experiment to monitor relative expression at very early stages after glyphosate application. q-PCR was performed using SYBR™ Select Mix (Invitrogen, Waltham, MA, USA) on a Step-One-Plus Real Time PCR system (Applied Biosystems, Waltham, MA, USA). The reactions were carried out with 2 μL of a four-fold dilution of cDNA, 10 μL SYBR™ Select Master Mix, 7.6 μL nuclease-free water, and 0.2 μL from each primer (0.4 μM). Primers of the four different *ABC-transporter* genes of interest were designed using the NCBI-primer tool and reported in Table 2.

**Table 2.** Primers used for the quantitative polymerase chain reaction (qPCR) analyses for the four different *ABC-transporter* genes of interest and reference gene.

| Primer Name | Primer Sequence |
| --- | --- |
| *LpABCIII-F* | 5′-AGAGCTGCAAAGGCTGGTAG-3′ |
| *LpABCIII-R* | 5′-TCTAAGCGGAAGCAAAGCCA-3′ |
| *Lpmultidrugprotein-F* | 5′-GGTCATGGACTGCGACAGAG-3′ |
| *Lpmultidrugprotein-R* | 5′-CACGTCAGATGACCGGTTTG-3′ |
| *LpM10-F* | 5′-TATGTTGTGGCTGACACGCT-3′ |
| *LpM10-R* | 5′-ATCGGCGTTGTGCAAGAAAT-3′ |
| *LpironinhibitedABC-F* | 5′-TAAACTCCCACCACCAGTGC-3′ |
| *LpironinhibitedABC-R* | 5′-TCACCGGTCATGAGCTTCAG-3′ |
| *Reference gene LpCCR-F* | 5′-GATGTCGAACCAGAAGCTCCA-3′ |
| *Reference gene LpCCR-R* | 5′-GCAGCTAGGGTTTCCTTGTCC-3′ |

qPCR conditions were as follows: hold temperature at 95 °C for 2 min, followed by 40 cycles: denaturation at 95 °C for 15 s; 60 °C for 1 min (annealing/extension). Finally, melting curve stage, for product quantification, were 95 °C for 15 s; 60 °C for 1 min and 95 °C for 15 s. Three individual biological replicates were assayed per qPCR run, in

three technical replicates. Relative expression of each gene was estimated using the ΔΔCt method [36]. Relative expression profiles for each target gene were estimated for all the time points studied for both treated plants and untread controls.

## 3. Results

### 3.1. Dose-Response Experiment

The relative fresh weight, at the recommended dose, ranged from 15% to 56% of the untreated control for each accession. At the highest herbicide dose, the relative biomass varied between 7% and 30% of the untreated (Figure 1). Not one of the populations was completely controlled at dose 1×, not even pop.3 collected in a field which had never been treated with glyphosate. The most susceptible was population 2, which was therefore used for the calculation of RI. Populations 1, 2 and 4 proved to be clearly resistant to glyphosate and were poorly controlled even at the highest dose (5760 g ae ha$^{-1}$) (Table 3). At intermediate doses, pop.4 collected in Italy was the most resistant.

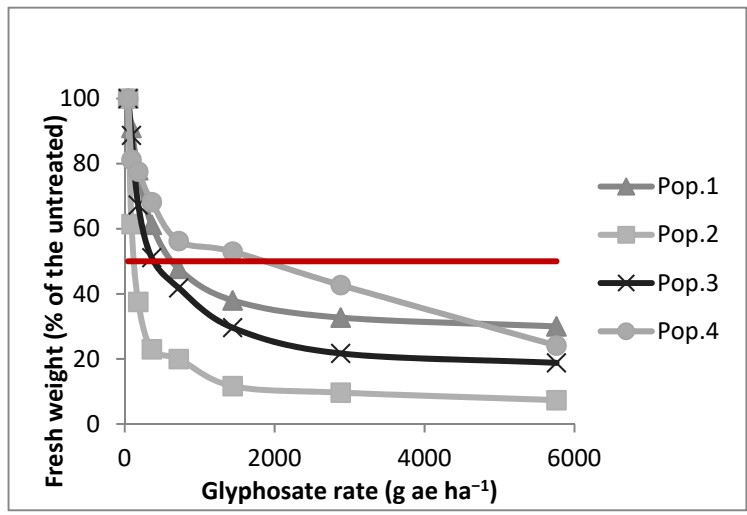

**Figure 1.** Fresh weight of *Lolium* spp. populations expressed as percentage (%) of the untreated control for different glyphosate rates at 4 weeks after treatment (WAT). Red line indicates the 50% of fresh weight. Populations 1, 3 and 4 were resistant to glyphosate whereas population 2 was the susceptible one.

**Table 3.** LD$_{50}$, LD$_{90}$, GR$_{50}$, GR$_{90}$ values (g ae ha$^{-1}$) for the studied populations. Standard errors are given in parentheses.

|  | Populations | | | |
|---|---|---|---|---|
|  | **Pop.1** | **Pop.2** | **Pop.3** | **Pop.4** |
| LD$_{50}$ | >5760 (60.18) | 270 (10.04) | >5760 (49.94) | 2850 (33.82) |
| LD$_{90}$ | >5760 (40.32) | 1580 (31.53) | >5760 (55.20) | >5760 (39.21) |
| GR$_{50}$ | 490 (19.05) | 126 (5.87) | 360 (13.08) | 740 (13.45) |
| GR$_{90}$ | >5760 (46.51) | 1240 (12.29) | >5760 (46.67) | >5760 (47.60) |

The RI for the three populations were around 5 (Table 4). Considering that pop.3, used as "susceptible" is not fully susceptible, it is likely that the RI are underestimated.

**Table 4.** Resistance indices for populations 1, 3 and 4, calculated as the ratio of the $GR_{90}$ of the considered R populations (pop.1, 3, 4) to the $GR_{90}$ of the considered S population (pop.2).

|  | $GR_{90\ (pop.1)}/GR_{90\ (pop.2)}$ | $GR_{90\ (pop.3)}/GR_{90\ (pop.2)}$ | $GR_{90\ (pop.4)}/GR_{90\ (pop.2)}$ |
|---|---|---|---|
| **RI** | 5.2 | 4.9 | 5.1 |

### 3.2. Shikimate Measurements

Results indicated that shikimate concentrations were significantly different among the studied populations. Population 2 accumulated the highest amount of shikimic acid, in contrast to population 1, where shikimate accumulation was almost half compared to population 2. Shikimate concentration for populations 3 and 4 was intermediate to populations 1 and 2. These measurements are consistent with dose-response results, which demonstrate that population 2 is the susceptible one and populations 1, 3 and 4 are clearly resistant (Figure 2).

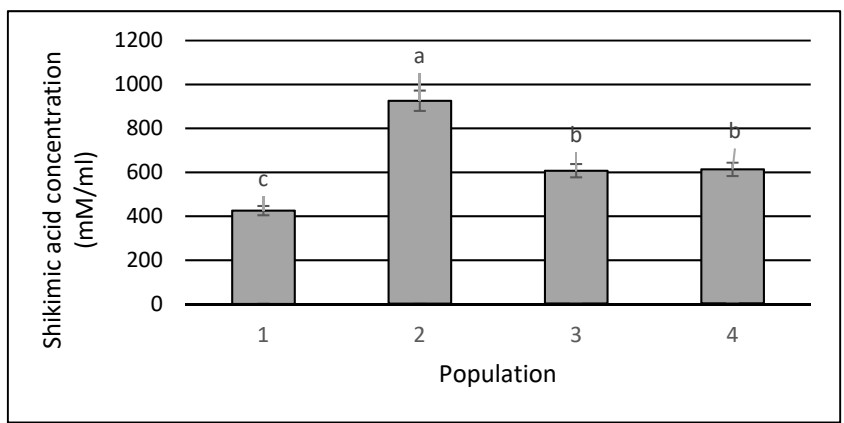

**Figure 2.** The different values of shikimic acid concentration for the four studied populations. Populations 1, 3 and 4 accumulated less shikimate acid and can be considered resistant to glyphosate in contrast to population 2, which is the susceptible one.

### 3.3. EPSPS cDNA Sequencing

A 564 bp fragment of the *EPSPS* gene of *L. rigidum* was retrieved from the NCBI database (MK492452.1) and compared to our three *Lolium* spp. populations. The sequences were aligned using the ClustalW aligning tool. The amino acid sequence of the *EPSPS* gene, which contains the proline 106 codon, is as follows:

FLGNAGTAMRPLTAAVVAAGGNATYVLDGVPRMRERPIGDLVVGLKQ GANVD-CFLGTDCPPVRINGIGGLPGGKV.

The sequencing results showed that most of the populations carry the proline amino acid encoded by the CCA codon (Table 4). However, population 4 and one plant sample of population 2 contained the proline amino acid encoded by CCG instead of CCA codon (Figure 3).

```
Pro 106
L. rigidum  TCTTCTTGGGGAACGCTGGAACTGCGATGCGGCCATTGACGGCTGCTGTAGTAGCTGCTG   60
Pop.1       TCTTCTTGGGGAACGCTGGAACTGCAATGCGGCCATTGACGGCAGCTGTAGTAGCTGCTG   60
Pop.2       TCTTCTTGGGGAACGCTGGAACTGCGATGCGGCCGTTGACGGCTGCTGTAGTAGCTGCTG   60
Pop.3       TCTTCTTGGGGAACRCTGGWACTGCGATGCGGCCATWGACGGCDGCTGCTGYARTARCTGCTG   60
Pop.4       TCTTCTTGGGGAACGCTGGAACTGCGATGCGGCCGTTGACGGCGGCTGTAGTAGCTGCTG   60
            *************** **** ***** ******** * ****** **** * ** ******
```

**Figure 3.** Nucleic acid sequences alignment of the 4 populations studied. CCA silent mutation is highlighted in yellow. CCG silent mutation is highlighted in green

In addition, a correlation seemed to exist between positions 106 and 109 in the amino acid sequence. The amino acid sequences in Table 5 indicates that for population 4, the silent mutation of proline at position 106 from CCA to CCG is also accompanied by a silent mutation of alanine at position 109 from GCT to GCG. For population 2 it was observed that in two out of the three samples analyzed, no change in the CCA codon was observed, while alanine codon 109 showed a silent point mutation from GCT to GCA. In the third sample of population 2, despite a silent proline mutation from CCA to CCG, alanine codon remained the same. No mutations in proline codon were detected for populations 1 and 3, however alanine is coded with the GCA and GCG codon instead of GCT in population 1. For population 3, sequencing did not give us a clear result for the alanine codon. In conclusion, no mutation conferring target site resistance to glyphosate was detected in any population at position proline 106.

**Table 5.** Proline (Pro 106) and alanine (Ala 109) codons of the sequenced population samples.

| Population | Pro 106 CCA | Ala 109 GCT |
|---|---|---|
| 1 s1 (R) | CCA | GCA |
| 1 s2 (R) | CCA | GCA |
| 1 s3 (R) | CCA | GCG |
| 1 s4 (R) | CCA | GCG |
| 2 s1(R) | CCA | GCA |
| 2 s2 (R) | CCG | GCT |
| 2 s3 (R) | CCA | GCA |
| 3 s1 (S) | CCA | - |
| 3 s2 (S) | CCA | - |
| 4 s1 (R) | CCG | GCG |
| 4 s2 (R) | CCG | GCG |

*3.4. Analysis of the Expression of ABC Transporter Genes*

Two Greek populations (*L. rigidum*, 1 and 3) and one Italian (*L. multiflorum* population (4) were tested using qPCR.

The relative transcript expression of the *ABC III gene* in population 4, 12 h after glyphosate treatment, was 60 times higher compared to 3 h after spraying. An increased relative expression at 12 h compared to 3 h after glyphosate application was also monitored for population 1. A gene up-regulation was monitored also for population 3, although at a lower rate, at 12 h after spraying (Figure 4).

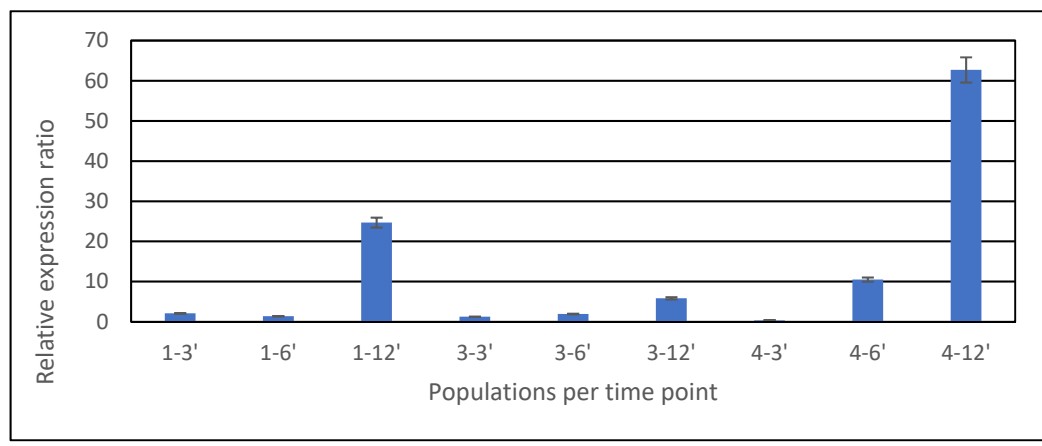

**Figure 4.** Relative expression ratio of the *ABC III* gene for the populations 1, 3 and 4 in glyphosate-applied leaves (3, 6, 12 h after glyphosate application).

Regarding the *Lpmultidrugptrotein* gene, population 4 presented a rapid response 3 h after glyphosate application followed by population 3, with an increased relative

expression 6 h after treatment. Population 1 showed a small increase (+2-fold change), 6 h after glyphosate application (Figure 5).

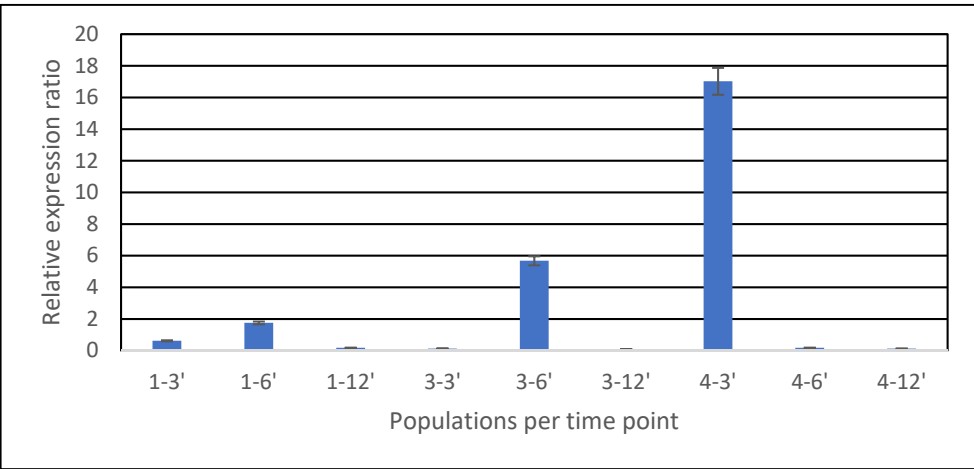

**Figure 5.** Relative expression ratio of the *Lpmultidrugprotein* gene for the populations 1, 3 and 4 in glyphosate-applied leaves (3, 6 and 12 h after glyphosate application).

Similarly, for the *Lpironinhibited ABC transporter* gene, population 4 responded 3 h after glyphosate application (eight-fold up-regulation), while population 3 showed a three-fold up regulation at 12 h after treatment. A two-fold up regulation was also observed for population 1 at 6 h after glyphosate application (Figure 6).

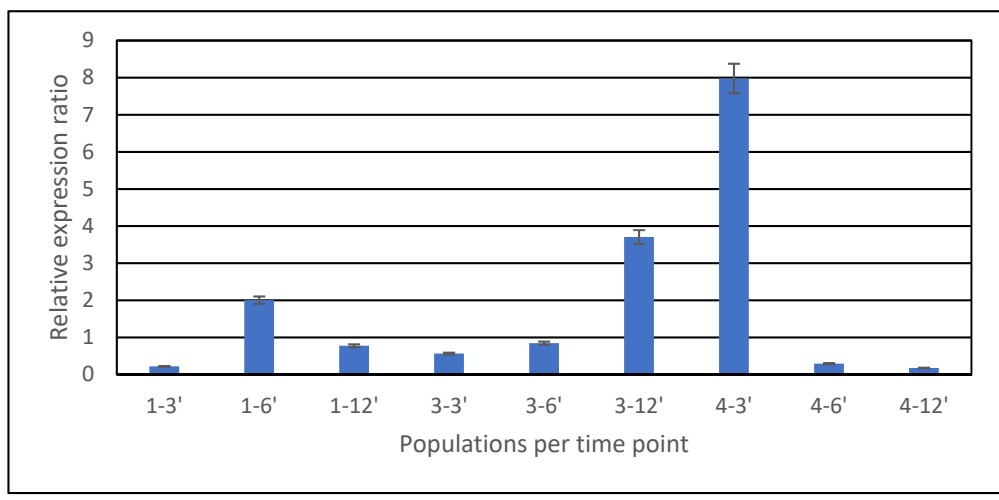

**Figure 6.** Relative expression ratio of the *LpironinhibitedABC* gene for the populations 1, 3 and 4 in glyphosate-applied leaves (3, 6 and 12 h after glyphosate application).

The mRNA accumulation of the *LpM10 transporter* gene increased rapidly in population 4 at 3 h after treatment. For this *ABC transporter* gene, populations 1 and 3 showed a small increase in relative transcript expression (two-fold change) at 3 and 6 h after treatment, respectively (Figure 7).

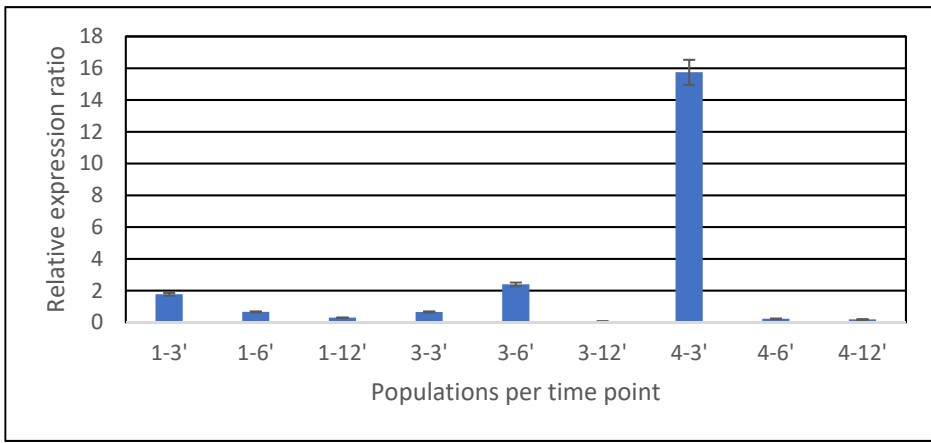

**Figure 7.** Relative expression ratio of the *LpM10* gene for the populations 1, 3 and 4 in glyphosate-applied leaves (3, 6 and 12 h after glyphosate application).

## 4. Discussion

As highlighted in a recent review [37], *Lolium* spp. hold many characteristics, (i.e., high genetic diversity and ability to exchange genetic material due to gene flow) that enable them to evolve resistance mechanisms to a plethora of different herbicides. On the other hand, glyphosate is one of the most efficient post-emergence herbicides, mainly in cropping systems targeting grasses [38]. For all these reasons, advancing our knowledge on TSR and NTSR mechanisms of *Lolium* spp. populations to glyphosate, and developing genetic markers that identify fields contaminated with glyphosate-resistant *Lolium* populations is of high importance. In the present study an effort has been made to unravel the mechanisms of glyphosate resistance for the three *Lolium* spp. studied populations. Thus, we tried to clarify whether *EPSPS* gene mutation at Pro 106 codon is responsible for the development of resistance or whether the *ABC transporter* genes are important in regulating these resistance mechanisms.

The estimated $LD_{50}$ and $GR_{50}$ values confirmed that populations 1, 3 and 4 can be considered as glyphosate resistant. A similar work presented by [39] in 2017, where two resistant populations and one susceptible population were studied, demonstrates that glyphosate $LD_{50}$ values were 93.21, 4280 and 2318 g ae ha$^{-1}$ for the susceptible and the two resistant populations, respectively, that are comparable to our results (Table 3). The shikimate analysis also revealed that populations 1, 3 and 4 can be considered as resistant to glyphosate. Population 1 accumulated almost two times more shikimic acid compared to populations 2–4 (Figure 2). Shikimic acid accumulation varies between two times to 35 times more in the susceptible populations compared to the resistant ones, depending on the different time point of sampling collection after glyphosate application [14,40–42].

The three resistant populations sequenced in the present study, carried the Pro 106 encoded by the CCA or CCG codon, and no mutation was observed, as reported in similar studies [12,35]. In our experiment a silent mutation in the Pro 106 codon was observed in the resistant populations 2 and 4 as well as a silent mutation in Ala 109 (Table 4). However, this phenomenon was also observed but in a susceptible population in another study [12]. Considering the many studies conducted, multiple independent mutation events seemed to take place in the EPSPS protein highlighting the complex evolutionary history of the target site resistance trait [43,44]. Often, glyphosate resistance caused by lower shikimate accumulation is associated with a greater *EPSPS* gene amplification or higher *EPSPS* transcript levels leading to an increased EPSPS protein expression [45]. In the present experiment, such studies were not conducted and, consequently, the only conclusion that can be drawn is that glyphosate resistance of these populations was not target-site based. In this work, the relative transcript levels of four *ABC transporter* genes (previously studied for their involvement to selenium tolerance, [34] were determined, at three different sampling times after glyphosate application. The increased relative transcript

levels of all the studied genes were detected at an extremely early stage after glyphosate application implying a rapid sequestration of glyphosate into vacuoles. Several studies also demonstrated overexpression of *ABC-transporter* genes after glyphosate application on *Conyza* spp. resistant populations, but this overexpression was observed 24 h after treatment [46]. In a similar study, *ABC transporters* exhibited significantly higher expression levels in the resistant *Conyza* spp. populations 24 h after glyphosate treatment [27]. By contrast, when we investigated the induction of these *ABC transporter* genes 24 h after glyphosate treatment in our *Lolium* spp. resistant populations, no accumulation of transcript levels was observed (data not shown). Our results indicated that *ABC transporter* genes are induced faster in *Lolium* spp. populations than in *Conyza* spp. and they can be a useful tool to rapidly identify (3 to 12 h after glyphosate application) the glyphosate-resistant populations of *Lolium* spp. In particular, the *ABC type III transporter* gene gave the most straightforward results (12 h after glyphosate treatments) and is a strong glyphosate resistance gene candidate that should be studied in more *Lolium* spp. resistant populations (Figure 4). The finding of our study can be an initial step towards creating a useful technique to rapidly identify *Lolium* spp. populations resistant to glyphosate. This tool would be useful for both researchers and farmers, who would have the chance to incorporate cropping practices more rapidly and efficiently in order to manage glyphosate resistance in *Lolium* spp.

**Author Contributions:** All authors contributed to the design of this study, shared plant materials, contributed to give comments, revised the manuscript, and read and approved the final manuscript. M.G. and N.C. performed the experiments and analyzed the data. E.T., M.G. and N.C. wrote the first draft of the manuscript. All authors have read and agreed to the published version of the manuscript.

**Funding:** This research (project RELIUM) was funded by the Italian Ministry of Agriculture and Forestry Politics (MIPAAF), the Danish Environmental Protection Agency (DEPA), and the Hellenic Agricultural Organisation (DEMETER) under the umbrella of the ERA-Net C-IPM (Coordinated Integrated Pest Management in Europe) funded by the European Union, Grant agreement No. 618110.

**Data Availability Statement:** The data sets generated and/or analyzed during the current study are available from the corresponding author on reasonable request.

**Acknowledgments:** The authors would like to thank Konstantinos Papadopoulos for helping with the *EPSPS* sequencing experiments.

**Conflicts of Interest:** The authors declare no conflict of interest.

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
