# Peer review of "Biochemical and Rapid Molecular Analyses to Identify Glyphosate Resistance in Lolium spp."

_agronomy, doi:10.3390/agronomy12010040_

Round 1

Reviewer 1 Report

The study is good and quite extensive. The major question remains that why only ABC transporter were studied.  I was also wondering how the expression patterns are for Cytochromes and GSTs? This is not clearly explained here.

Author Response

We would like to thank the reviewer for their helpful comments. The involvement of ABC transporter genes in glyphosate sequestration into vacuoles has been implicated in several studies (Shaner 2009; Tani et.al. 2015; Ghanizadeh  and Harrington 2017). After herbicide application, findings reveal that ABC transporter genes reduce plant uptake and herbicide movement within the plant, leading to retention of glyphosate at the initial point of application (De Prado et.al. 2005; Vila-Aiub et.al. 2012), transportation and isolation of the herbicide in vacuoles, where glyphosate remains inactive (Ge et.al. 2010) and finally rapid metabolism of the herbicide to non-harmful compounds (Busi et.al. 2011; De Carvalho et.al. 2012; De Prado et.al. 2005; Gonzalez-Torralva et.al. 2012). In our study we tried to find a rapid tool to identify populations with resistance to glyphosate and ABC transporters  are the most promising genes towards this goal.

Reviewer 2 Report

I have a few questions about the methodology. 

1) Why did the authors choose to study pro 106 codon mutation why not any other mutation?

2) Why authors have not sequenced the gene pro 106 codon without converting it to cDNA? 

Furthermore, I think that authors should clearly state, why they
are using two different species of Lolium (without including
sensitive standard). Throughout the text they compare them together
without telling that to the reader. Honestly, I think that these
are two different species, having partly different genetic background
and this can affect the results. I recommend to distinguish between L.
rigidum and L. multiflorum and possibly conclude, if there can be any
difference with a respect to results they obtained.

Author Response

We would realy like to thank the reviewer for their comments. 

1)The mutation at position 106 of proline is the most common mutation identified for TSR to glyphosate according to similar studies (Ng et.al. 2003; Fernandez et.al. 2015). In our study we wanted to identify, among others, whether the resistance mechanisms of glyphosate resistance in Lolium sp. are target site or non-target site based, so the most reliable point mutation position, in this respect, was Pro 106.

2)cDNA synthesis leads to a much more stable molecule in contrast to RNA. cDNA products are more stable both for treatment during the experimental procedure and transportation, as the cDNA samples sequenced from off-campus service. Additionally, many similar studies have treated their samples in a similar way (Jasieniuk et.al. 2008; Li et.al. 2012; Fernandez et.al. 2015).

3)In fact, it is not always clear whether we have the one or the other species in all cases since under real-field conditions 2-3 Lolium species sometimes co-exist. Cross pollination and hybridization do play a role here and therefore we prefer to keep it as Lolium spp. since in that way it is meaningful for the farmers as well. Furthermore, please also note that many papers in the past have similar approaches (without defining the species) like Singh et.al. 2020; Panozzo et.al. 2020; Scarabel et.al. 2021 etc. Therefore, we prefer keeping it in that way. In our study we tried to find a tool for rapid identification of resistant Lolium sp. populations.